# Antigiardial Activity of *Foeniculum vulgare* Hexane Extract and Some of Its Constituents

**DOI:** 10.3390/plants11172212

**Published:** 2022-08-26

**Authors:** Irma G. Domínguez-Vigil, Benito D. Mata-Cárdenas, Patricia C. Esquivel-Ferriño, Francisco G. Avalos-Alanís, Javier Vargas-Villarreal, María del Rayo Camacho-Corona

**Affiliations:** 1Facultad de Ciencias Químicas, Universidad Autónoma de Nuevo León, San Nicolás de los Garza 64570, NL, Mexico; 2Department of Therapeutic Radiology, Yale University School of Medicine, New Haven, CT 06510, USA; 3Laboratorio de Bioquímica y Fisiología Celular, Centro de Investigación Biomédica del Noreste, Instituto Mexicano del Seguro Social, Monterrey 64720, NL, Mexico

**Keywords:** *Foeniculum vulgare*, *Giardia duodenalis*, antigiardial activity

## Abstract

*Foeniculum vulgare* is used for the treatment of diarrhea in Mexican traditional medicine. Hexane extract showed 94 % inhibition of *Giardia duodenalis* trophozoites at 300 μg/mL. Therefore, 20 constituents of hexane extract were evaluated to determine their antigiardial activity. Interestingly, six compounds showed good activity toward the parasite. These compounds were (1*R*,4*S*) (+)-Camphene (61%), (*R*)(−)-Carvone (66%), estragole (49%), *p*-anisaldehyde (67%), 1,3-benzenediol (56%), and trans, trans-2,4-undecadienal (97%). The aldehyde trans, trans-2,4-undecadienal was the most active compound with an IC_50_ value of 72.11 µg/mL against *G. duodenalis* trophozoites. This aldehyde was less toxic (IC_50_ 588.8 µg/mL) than positive control metronidazole (IC_50_ 83.5 µg/mL) against Vero cells. The above results could support the use of *F. vulgare* in Mexican traditional medicine.

## 1. Introduction

*Giardia duodenalis* (syn. *Giardia lamblia, Giardia intestinalis*) is a flagellated enteric protozoan and the etiologic agent of giardiasis, a relevant public health problem. *G. duodenalis* is the most common protozoal infection in humans and is an important cause of waterborne and foodborne diarrhea, day-care center outbreaks, and traveler’s diarrhea [1]. It contributes to an estimated 280 million human cases of diarrhea every year and has been included as part of the WHO Neglected Disease Initiative associated with poverty [1,2,3]. The prevalence of giardiasis in humans ranges from 2–3% in industrialized countries, up to 30% in low-income and developing countries [4].

The life cycle of *G. duodenalis* is direct and involves two stages, the trophozoite, which is the replicative stage, and the cyst, which is the infective stage [5]. Giardiasis mainly occurs in children, and a serious manifestation is its contribution to malnutrition and the stunting of growth following early or repeated infections [6]. It is spread among people with poor hygiene habits, and may result in asymptomatic, acute, or chronic disease, depending on multiple factors such as the age of the patient, history of prior exposure, the parasite load, virulence of the parasite, and the immune response of the host [1]. Beyond diarrhea, symptoms include abdominal bloating and cramps, malabsorption and weight loss [2].

Adequate sanitation is not sufficient for protection against giardiasis and there are no available biological prophylactic methods against *G. duodenalis* infections, meaning that its control is reliant on treatment [3].

There are numerous commercially available drugs for the treatment of giardiasis including nitroimidazole derivatives (metronidazole, tinidazole, secnidazole and ornidazole), benzimidazoles (albendazole, mebendazole), nitazoxanide, furazolidone, quinacrine, chloroquine and paromomycin [6]. Although, there are many antigiardial drugs available, only a few are used in daily practice. In 90% of cases, treatment regimens with a single agent, either metronidazole or albendazole, are effective, but drug resistance has been reported [7,8], leading to combinations of the therapies.

In addition to drug resistance, other factors can also be responsible for treatment failure, such as reinfection in endemic areas with poor sanitation and hygiene, insufficient amounts of drug administered, immunosuppression, or sequestration in the gallbladder or the pancreas [9]. Adverse effects such as nausea, abdominal pain, and diarrhea have also been associated with the use of these drugs, perhaps most controversial is their potential to be genotoxic and carcinogenic in humans [3,10]. Therefore, there is a need to develop new, safer antigiardial agents, and natural products could be an excellent alternative, particularly in communities with limited access to health services.

*Foeniculum vulgare* Mill. (Apiaceae family), commonly known as fennel, is a widespread perennial plant with an aromatic odor. It is native to Southern Europe and the Mediterranean region and is currently widely cultivated throughout the temperate and tropical regions of the world. The chemical constituents of fennel include fatty acids, phenylpropanoids, terpenes, coumarins, flavonoids, and other types of compounds [11].

The essential oil of fennel fruits is used for flavoring purposes, and in cosmetic and pharmaceutical products. Extracts of *F. vulgare* and some of its constituents have shown antioxidant, antibacterial, antifungal, estrogenic, and antituberculosis activities [11]. Aerial parts of *F. vulgare* are used in Mexico for the treatment of bronchitis, coughs, dysmenorrhea, diarrhea, and abdominal pains [12]. The aim of this study was to determine the antigiardial activity of *F. vulgare* hexane extract and some of its components to support its use in the traditional Mexican medicine.

## 2. Results

Hexane extract of *F. vulgare* was prepared previously [13], and tested for antigiardial activity, showing 94% inhibition of *G. duodenalis* trophozoites at 300 μg/mL. The chemical composition of this extract was determined using gas chromatography-mass spectrometry (GG-MS) by our research group, identifying the compounds by their mass spectra fragmentation patterns using the NIST 1.7 library database [13].

Nineteen compounds identified previously in the hexane extract [13] were tested for antigiardial activity at 300 μg/mL. Results in Table 1 show that six compounds, including the bicyclic monoterpene, (1*R*,4*S*)(+)-camphene (61%); the monoterpene, (*R*)(−)-carvone (66%); the aromatic ether, estragole (49%); the aromatic aldehyde, *p*-anisaldehyde (67%); the diphenol, 1,3-benzenediol (56%); and the aldehyde, *rans*, *trans*-2,4-undecadienal (97%), inhibited *G. duodenalis* trophozoites in the range of 49 to 97%, with the aldehyde *trans, trans*-2,4-undecadienal being the most active compound. The above activity of the monoterpenes (1*R*,4*S*) (+) camphene was shown by one stereoisomer; however, it is necessary to determine the antigiardial activity of the other stereoisomers to see if there is a structure–activity relationship.

On the other hand, the activity of the monoterpene (*R*) (−)-carvone was given by one enantiomer, so it will be necessary to examine the other enantiomer. For example, the antigiardial activity of the enantiomers (*S*) (+)-fenchone and (*R*)(−)-fenchone showed 16 and 6.5 % of inhibition at 300 µg/mL, respectively. It can be seen that tructure–activity relationship may be reflected in their antigiardial activity because the first terpene was 2.46-times more active than the second one.

Lastly, we have not anticipated if there is any difference in the antigiardial activity of the *o*-, *m*, or *p*-cymenes. Thus, it will be necessary to carry out these experiments in the future.

In addition, half-maximal inhibitory concentrations (IC_50_) towards *G. duodenalis* trophozoites and Vero cells were determined for the hexane extract, as well as for the six most active compounds that showed a percentage of inhibition (≥49%) of trophozoites (Table 2). These were all compared to metronidazole, the most common first choice of treatment against *G. duodenalis*.

The most active compound against *G. duodenalis* was the aldehyde *trans*, *trans*-2,4-undecadienal (IC_50_ 72.1 μg/mL) as seen in Table 2. However, it was not as active as the positive control metronidazole (IC_50_ 0.5 μg/mL) against the parasite. Interestingly, the mentioned aldehyde was less toxic (IC_50_ 588.8 μg/mL) than metronidazole (IC_50_ 83.5 μg/mL) against Vero cells.

Additionally, we determined the selectivity index (SI) of the most active compounds against *G. duodenalis*. SI values ≥ 1 are considered promising candidates against parasitic infection. In our study, the six most active compounds of *F. vulgare* could be considered as potential candidates for the treatment of giardiasis.

## 3. Discussions

Esquivel-Ferriño et al. identified the aldehyde *trans, trans*-2,4-undecadienal in the hexane extract of *F. vulgare* and reported its activity against the sensitive strain H37Rv (MIC 25 μg/mL) and three multidrug resistant (MDR) clinical isolates (MIC 25–50 μg/mL) of *Mycobacterium tuberculosis* [13]. This aldehyde has also been identified by GC-MS from coriander (*Coriandrum sativum* L.) and found to be a highly effective deodorant compound against the offensive odor of the porcine large intestine [14].

It is worth mentioning that unsaturated aldehyde *trans*, *trans*-2,4-undecadienal is lipophilic in nature (log P 3.8) in contrast to the positive control metronidazole which is hydrophilic in nature (log P 0.1), its structure is seen in Figure 1. Therefore, it is possible that they have different mechanisms of action. It is well known that metronidazole forms covalent adducts with cysteines in proteins. Specifically, metronidazole targets the redox enzyme thioredoxin reductase in *Entamoeba histolytica*, *Trichomonas vaginalis*, and *G. duodenalis* and inhibits the enzyme’s function as a disulfide reductase, resulting in severe oxidative stress [15,16,17].

As constituents of plants, aldehydes play an important role in antimicrobial activity [18]. Aldehydes react with biologically nucleophile groups. In particular, *α*,*β*-unsaturated aldehydes are known to react with sulfhydryl, amino and hydroxyl groups under physiological conditions. The mode of action of the aldehyde *trans, trans*-2,4-undecadienal is likely due to its penetration in the outer layer of the parasite and the alteration of the function of membrane-associated proteins in its cellular surface causing its destabilization and death.

In addition, it has been reported that aldehydes such as *trans*, *trans*-2,4-decadienal are metabolites of lipid peroxidation decomposition. These are able to form adducts of type 1, N^6^-etheno-2’-deoxyadenosine, producing direct damage to DNA, the cell wall and membrane, and generating the activation of antioxidant factors such as GSH, cytochrome C and protein stress to hypoxia [19,20,21].

Brandelli et al. have shown that the aqueous extract of aerial parts of *F. vulgare* was inactive against the trophozoites of *G. duodenalis* [22]. In contrast, we were able to show inhibition against *G. duodenalis* because the hexane extract has lipophilic constituents whereas the aqueous extract has hydrophilic compounds. It is the lipophilic nature of these compounds that seems to be responsible for the antigiardial activity of *F. vulgare*.

The essential oil of *F. vulgare* has been tested against the helminth *Schistosoma mansoni*, exhibiting moderate in vitro antiparasitic activity on its adult worm form [23].

Moreover, *F. vulgare* aqueous extract was tested against the parasitic protozoa *Blastocystis* spp. showing a dose- and time-dependent anti-Blastocystis activity at 250 µg/mL [24].

Finally, Karami et al., have tested the efficacy of the extracts, essential oil, and trans-anethole (its main essential oil component) of *F. vulgare* against the protozoan parasite *Trichomonas vaginalis* suggesting an in vitro antiprotozoal property [25]. 

Our results support the use of *F. vulgare* as an antidiarrheal treatment in Mexican traditional medicine. Additionally, the compound *trans, trans*-2,4-undecadienal showed a promising toxicity effect toward *G. duodenalis*. This aldehyde could serve as a basic structure for future modifications in the search of new antiprotozoal drugs.

To our knowledge, this is the first report of the antigiardial activity of *F. vulgare* and some of its constituents. In addition, in vivo assays of the most active compounds as well as the elucidation of their mechanisms of action should be performed.

## 4. Materials and Methods

### 4.1. Plant Material and Extraction 

Stems and leaves of *F. vulgare* var. *dulce* were collected in Pesquería, Nuevo León (México) in July 2017. Plant material was provided and identified by biologist Mauricio González Ferrara; a sample with a voucher number 024771 was deposited at the herbarium of the Department of Botany, Universidad Autónoma of Nuevo León, Mexico. Powdered air-dried leaves and stems (944 g) were extracted by maceration a at room temperature using 5 L of hexane once. Extract was filtrated and distilled under vacuum yielding 6.2 g of hexane extract [13].

### 4.2. Chemicals

The compounds identified in the hexane extract by GC-MS were purchased from Sigma-Aldrich (St. Louis, MO, USA). These were: fenchylacetate, *α*-tujone, (*R*) (−)-carvone, *o*-cymene, *p*-anisaldehyde, (*R*)(−)-fenchone, (*S*)(+)-fenchone, terpinolene, methylchavicol, trans-anethol, estragole, eugenol, 1,3-benzenediol, undecanal, (*R*)-(+)-*β*-citronellol, *trans, trans*- 2,4-undecadienal, (1*R*, 4*S*)(+)-camphene, and oleic acid. The solvents used for the extractions were purchased from J.T. Baker (USA).

### 4.3. Parasite Culture

*G. duodenalis* strain 0989: IMSS was used. *G. duodenalis* was cultured in TYI-S-33 media supplemented with bile, as previously described [26]. *G. duodenalis* trophozoites were sub-cultured three times a week. Parasites used in the assays to determine drug susceptibility were harvested when cultures reached the middle of their logarithmic growth phase.

### 4.4. Cytotoxic Activity on Vero Cell Line

Cytotoxicity assay was performed on Vero cell line (ATCC CCL-81), as previously described [27]. Briefly, Vero cells were cultivated in Roswell Park Memorial Institute (RPMI) medium supplemented with 10% fetal bovine serum (FBS). Only the most active compounds were tested. Each compound was dissolved in dimethyl sulfoxide (DMSO) and sterilized by filtration, and brought to final concentrations of 487, 243.5, 121.75 and 60.875 μg/mL with RPMI medium supplemented with 10% FBS. The cell suspension was adjusted to 1 × 10^4^ cells/mL with RPMI medium supplemented with 10% FBS. Cell suspension (200 μL) was placed into a 96-well plate and incubated at 37 °C in a 5% CO_2_ atmosphere for 24 h. Each compound solution was added to each well containing the Vero cells and incubated for 24 h. The assay was carried out in triplicate. The number of cells were determined in a Neubauer chamber, while the percentage of viable cells was measured with the Trypan blue staining method. The half-maximum inhibitory concentration (IC_50_) of each drug was calculated by Probit method.

Additionally, selectivity index (SI) for the compounds was determined to identify the quantity of the compound that is active against the parasite with low toxicity to the host cells. The SI is the ratio of the IC_50_ concentration in host cells (Vero cells) to that in parasite (*G. duodenalis*) [28].

### 4.5. Antigiardial Activity

The concentration of all standard drugs was adjusted to 10 mg/mL. Metronidazole (used as a positive control), and pure compounds were dissolved in (DMSO). All stock solutions were stored at –20 °C until use. Before use in the assays, serial two-fold dilutions of the stock solutions were made in basal TYI medium (without serum). Fifty micro liters of each solution was added into 1 mL glass screw-capped cylindrical vials with a conical interior (Wheaton vial). All vials were filled with 950 μL of a freshly prepared parasite suspension in TYI-S-33 medium plus 10 % bovine serum. *G. duodenalis* was tested at density of 2 × 10^5^ trophozoites/mL. All vials were incubated at 36 °C for 24 h. The vials were then chilled on iced water for 20 min, and the number of trophozoites per milliliter in each tube were counted using a hemocytometer (Neubauer cell-counter chamber) [29]. The percentage of growth inhibition with respect to untreated controls was then determined.

### 4.6. Statistical Analysis

All experiments were performed in triplicate and in at least three independent bioassays (n = 9 cultures for each analyzed compound). The data were analyzed using GraphPad Prism software version 5.0 (USA). The half-maximum inhibitory concentration (IC_50_) of each drug was calculated by Probit analysis and 95% confidence limits were calculated.

## 5. Conclusions

The results suggest the in vitro antigiardial properties of *F. vulgare* hexane extract. Antigiardial properties of *F. vulgare* are mainly due to the aldehyde *trans, trans*-2,4-undecadienal found in the hexane extract of this plant. In addition, it is possible that the other compounds contribute to the antigiardial activity or act in a synergist way. The results found in this study justify the use of *F. vulgare* for the treatment of diarrhea associated to giardiasis in Mexican traditional medicine.

## Figures and Tables

**Figure 1 plants-11-02212-f001:**
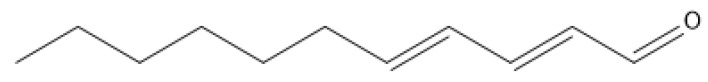
Structure of *trans, trans*-2,4-undecadienal.

**Table 1 plants-11-02212-t001:** Antigiardial activity of *F. vulgare* hexane extract and its chemical constituents.

Extract/Compound	% Inhibition at 300 μg/mL
Hexane extract	94
*Trans*, *trans*-2,4-undecadienal ^c^	97
*p*-Anisaldehyde ^b^	67
(*R*) (−)-Carvone ^a^	66
(1*R*,4*S*) (+)-Camphene ^a^	61
1,3-Benzenediol ^b^	56
Estragole ^b^	49
Fenchylacetate ^a^	38.5
(*R*)- (+)-β-Citronellol ^a^	38
*o*-Cymene ^b^	30
Terpinolene ^a^	30
Undecanal ^c^	25
*γ*-Terpinene ^a^	24
Oleic acid ^d^	24
Methylchavicol ^b^	20
*Trans*-anethol ^b^	17
(*S*) (+)-Fenchone ^a^	16
(−)-α-Tujone ^a^	14
(*R*) (−)-Fenchone ^a^	6.5
Pinacol ^e^	2
Metronidazol	100

^a^ terpene, ^b^ aromatic compound, ^c^ aldehyde, ^d^ fatty acid, ^e^ diol.

**Table 2 plants-11-02212-t002:** Antigiardial and cytotoxic activities of *F. vulgare* hexane and some of its constituents.

Extract/Compound/Drug	*G. duodenalis* IC_50_ (μg/mL) (IC_95%_)	Vero Cells IC_50_ (μg/mL) (IC_95%_)	Selectivity Index (SI)
Hexane extract	89.3 (66.6–116.7)	nd	nd
(1*R*,4*S*) (+)-Camphene	181.1 (158.1–207.9)	494.3 (492.1–496.4)	2.7
(*R*) (−)-Carvone	207.0 (162.2–267.3)	350.7 (348.4–353.0)	1.6
Estragole	150.0 (130.8–175.9)	258.7 (245–265)	1.7
*p*-Anisaldehyde	196.8 (165.6–234.4)	540.7 (538.6–542.9)	2.7
1,3-Benzenediol	168.7 (140.5–182.5)	437.90 (410.7–470)	2.5
*Trans*, *trans*-2,4-undecadienal	72.1 (57.7–90.1)	588.8 (586.7–591.0)	8.1
Metronidazole	0.5 (0.4–0.6)	83.5 (81–86.1)	160.2

nd: not determined.

## Data Availability

Not applicable.

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
