# Peer review of "Antigiardial Activity of Foeniculum vulgare Hexane Extract and Some of Its Constituents"

_plants, 2022, doi:10.3390/plants11172212_

Round 1

Reviewer 1 Report

Dear authors,

I have read your paper and it is all around well written and referenced. I did find some parts underwritten, and I would urge you to improve upon them. I will list both technical and scientific issues as follows:

1. When using Latin binomial names, please also cite the author of the taxa (both, species and genera)

2. Giardiasis is the name of the disease, so no italic formatting is needed

3. On line 58, you are missing an "as" after "Adverse effects such..."

4.  Fennel is a perennial, so please correct the statement on line 64

5. On line 65, please change the wording to either "Mediterranean region" or "certain regions of the Mediterranean"

6. In Table 1, please order the compounds either by inhibition level or alphabetically

7. On line 104 I believe you are referring to Table 2 instead of Table 3.

8. In Table 2, please widen the first column so all rows have the same height. You have a fairly small table so it shouldn't be a problem. I would suggest depicting the results presented in Table 2 as a graph (bar chart).

9. On line 117 there is no need to mention the year of the reference in the brackets due to the citation style used.

10. On lines 124 and 125 please change the text to "is lipophilic/hydrophilic in nature"

11. When referencing authors in text, please don't use italic formatting on et al., on the other hand, please use italic formatting when using terms like in vivo, in vitro

12. On line 165 you are mentioning a variety, so please use italic formatting for "dulce"

13. Please provide a more detailed description of the extraction procedure

14. Please define abbreviations like RPMI and DMSO upon the first mention

15. Why didn't you use negative control (untreated cells) as well? The data could be used to improve the statistical part of your research and strengthen your conclusions.

16. I would advise using ANOVA with some sort of post hoc test in order to test differences in cytotoxic and antigiardial activity of each compound compared to positive control.

Author Response

Reviewer:1 Comments and Suggestions for Authors Dear authors, I have read your paper and it is all around well written and referenced. I did find some parts underwritten, and I would urge you to improve upon them. I will list both technical and scientific issues as follows:

  1. When using Latin binomial names, please also cite the author of the taxa (both, species and genera) (We have used both binomial names for the parasite, including the species and genera, see lines 36 and 46).
  2.  Giardiasis is the name of the disease, so no italic formatting is needed (We edited the sentence, see line 45).
  3. On line 58, you are missing an "as" after "Adverse effects such..." (We have edited the sentence, see line 58).
  4.  Fennel is a perennial, so please correct the statement on line 64 (We edited the sentence, see line 64).
  5.  On line 65, please change the wording to either "Mediterranean region" or "certain regions of the Mediterranean" (We edited the sentence, see lines 64-65).
  6.  In Table 1, please order the compounds either by inhibition level or alphabetically (We have organized the table 1 by inhibition level, see Table 1, line 103).
  7.  On line 104 I believe you are referring to Table 2 instead of Table 3. (We edited the sentence, see line 113).
  8.  In Table 2, please widen the first column so all rows have the same height. You have a fairly small table so it shouldn't be a problem. I would suggest depicting the results presented in Table 2 as a graph (bar chart). (We edited the Table 2, see line 122. We thank for the valuable suggestion to present 2 our results as a bar chart; however, we have decided to present our results as a table as we have three different analyses, and we consider it would be easier to compare them in a table format).
  9.  On line 117 there is no need to mention the year of the reference in the brackets due to the citation style used. (We edited the sentence, see line 126).
  10.  On lines 124 and 125 please change the text to "is lipophilic/hydrophilic in nature" (We edited the sentence, see lines 132-134).
  11.  When referencing authors in text, please don't use italic formatting on et al., on the other hand, please use italic formatting when using terms like in vivo, in vitro (We edited the sentence, see lines 126,155, and 164 for et al. formatting; see line 166 and 172 for in vitro and in vivo formatting)
  12.  On line 165 you are mentioning a variety, so please use italic formatting for "dulce" (We edited the sentence, see line 177).
  13.  Please provide a more detailed description of the extraction procedure (We have provided a more detailed description of the extraction procedure, see lines 181-183).
  14.  Please define abbreviations like RPMI and DMSO upon the first mention (We have defined the abbreviations, see lines 202 and 204).
  15.  Why didn't you use negative control (untreated cells) as well? The data could be used to improve the statistical part of your research and strengthen your conclusions. (We appreciate this very good question. We indeed used a negative control for our experiments, cells in presence of DMSO; DMSO was used as diluent of the pure compounds. DMSO did not affect the viability of our cultures and therefore was used as 100% viability. Since this is an internal control, we have decided not to show in our current tables)
  16.  I would advise using ANOVA with some sort of post hoc test in order to test differences in cytotoxic and antigiardial activity of each compound compared to positive control. (We really appreciate this very valuable suggestion; however, since Metronidazole, our positive control, has a very noticeable activity against the parasite, the difference between the antigiardial activity of the pure compounds versus the positive control would be too high, therefore we decided to keep our results as shown in Table 2).Please, enclose you will find a letter with our responses

Reviewer 2 Report

Chiral isomer mentioning is not clear.

Compound identification techniques is not clear.

Percentages only from GCMS is not correct. So I request author to rework on chemistry side. 

Author Response

Please, enclose you will find a letter with our responses

Reviewer 3 Report

Add the structure of most active compound. 

check the grammer of the line number 137

the improved activity of most active compound with respect to the extract can discuss in the discussion of paper

Author Response

Please, enclose you will find our letter with our responses

Round 2

Reviewer 1 Report

The authors have addressed majority of my questions raised. Therefore, I think that the manuscript is of good quality to be accepted for publication.